# Large-Scale Production of Anti-RNase A VHH Expressed in *pyrG* Auxotrophic *Aspergillus oryzae*

Elif Karaman [1,2], Alp Ertunga Eyüpoğlu [3], Lena Mahmoudi Azar [1] and Serdar Uysal [1,*]

[1] Beykoz Institute of Life Sciences and Biotechnology, Bezmialem Vakif University, 34820 Istanbul, Turkey; ekaraman@bezmialem.edu.tr (E.K.)
[2] Department of Biotechnology, Institute of Health Sciences, Bezmialem Vakif University, 34093 Istanbul, Turkey
[3] Department of Molecular Biology and Genetics, Faculty of Arts and Sciences, Bogazici University, 34450 Istanbul, Turkey
[*] Correspondence: suysal@bezmialem.edu.tr

**Abstract:** Nanobodies, also referred to as VHH antibodies, are the smallest fragments of naturally produced camelid antibodies and are ideal affinity reagents due to their remarkable properties. They are considered an alternative to monoclonal antibodies (mAbs) with potential utility in imaging, diagnostic, and other biotechnological applications given the difficulties associated with mAb expression. *Aspergillus oryzae (A. oryzae)* is a potential system for the large-scale expression and production of functional VHH antibodies that can be used to meet the demand for affinity reagents. In this study, anti-RNase A VHH was expressed under the control of the glucoamylase promoter in *pyrG* auxotrophic *A. oryzae* grown in a fermenter. The feature of *pyrG* auxotrophy, selected for the construction of a stable and efficient platform, was established using homologous recombination. Pull-down assay, size exclusion chromatography, and surface plasmon resonance were used to confirm the binding specificity of anti-RNase A VHH to RNase A. The affinity of anti-RNase A VHH was nearly 18.3-fold higher (1.9 nM) when expressed in *pyrG* auxotrophic *A. oryzae* rather than in *Escherichia coli*. This demonstrates that *pyrG* auxotrophic *A. oryzae* is a practical, industrially scalable, and promising biotechnological platform for the large-scale production of functional VHH antibodies with high binding activity.

**Keywords:** VHH; nanobodies; single domain antibodies; *Aspergillus oryzae*; *pyrG*; surface plasmon resonance; affinity reagent



## 1. Introduction

Heavy-chain antibodies (HCAbs) are functional antibodies consisting of a single variable domain on a heavy chain (VHH) and have been found in sera from species of the family Camelidae [1]. VHH antibodies, also known as nanobodies or single-domain antibodies, are smaller than the heavy chains of monoclonal antibodies (mAbs) because VHH lacks constant domain-1 (CH1), which normally ensures that HCAbs lacking the light chains can be secreted from the endoplasmic reticulum [2,3]. This disadvantage is compensated for by differences in the complementarity-determining regions (CDRs) and by the replacement of hydrophobic amino acids in the heavy chains with smaller hydrophilic amino acids in the VHH, which increases solubility and prevents aggregation in the absence of light chains [2].

In contrast to mAbs, an extra disulfide bond between CDR3 and CDR1/scaffold 2 provides stability to VHH antibodies, and an extended CDR3 domain endows VHH antibodies with highly specific antigen-binding affinity through high variability [2,4]. Because the extended CDR3 domain increases the binding surface area, VHHs can reach buried targets that mAbs cannot reach by producing a convex structure to bind a concave antigen with high specificity [2,5,6]. In addition, the high sequence homology (>80%) in the scaffold regions with human VH, the small size (4 nm in height and 2.5 nm in

diameter), hydrophilicity, and the absence of an Fc component all contribute to the low immunogenicity of VHH antibodies [7]. Furthermore, VHH antibodies exhibit a remarkable shelf life (of months at 4 °C or longer at −20 °C), without any decrease in activity [7], as well as resilience against harsh conditions, such as high temperatures [8], or in the presence of denaturants thanks to their capacity for refolding [9,10].

VHH antibodies are ideal affinity reagents that can be used for diagnostic, imaging, and therapeutic purposes due to their exceptional characteristics, such as small size and high solubility, specificity and affinity, tissue penetration, and stability [11–14]. Due to their specificity, flexibility, and ease of adaptability, VHH antibodies can be used as crystallization partners to determine protein conformation by stabilizing the intrinsic flexible region and increasing the surface area for contact with solvents [15,16]. They can also be used to function as co-purification partners [17], toxin neutralizers [18,19], or imaging tools [20,21]. VHH antibodies are rapidly eliminated from the body because of their small size, which is advantageous for visualizing tumor cells and their surroundings [21,22]. VHH antibodies can be used as targeted soluble probes or intracellular antibodies to monitor the subcellular localization of proteins in living cells with improved sensitivity and specificity [23,24]. Due to their stability, affinity, solubility, and simplicity of expression, VHH antibodies can also be exploited for therapeutic applications, such as the generation of CAR-T cells [25,26] and bispecific antibodies [27]. Caplacizumab (approved) [28], VHH 203027 (phase 2) [29], and ALX-0061 (phase 2) [30] are examples of therapeutic VHHs that have successfully completed clinical trials.

VHH antibodies are well suited for heterologous expression to achieve a high yield because optimal folding and stability are ensured due to their small size, hydrophilicity, strictly monomeric nature, and low complexity [31]. Because of their large (150 kDa) and complex heterodimeric structure, mAbs used as affinity reagents in imaging, diagnostic, or therapeutic applications can be expressed only in limited quantities compared with VHHs [32]. The main obstacles to the large-scale production of mAbs are low yield, expensive and time-consuming methods for mammalian expression, and issues with solubility and folding for bacterial expression [32–34]. Although the use of antigen-binding fragments (Fab) and single-chain variable fragments (scFv) allows for the reduction in the size of the recombinant protein expressed in host microorganisms, their heterologous expression is challenging because of domain mismatch, a propensity to aggregate, and low expression levels [35].

*Escherichia coli* (*E. coli*) periplasmic expression is the system most often used for VHH antibody expression, and VHH antibodies have been produced at various levels by *E. coli* [36–38]. However, limited periplasmic volume and insufficiency of chaperones are barriers in achieving a high yield of VHH antibodies [39]. *E. coli* cytoplasmic expression, however, is better suited for fusion techniques that increase the final yield. Although the use of fused VHH antibodies may necessitate a challenging purification process for the removal of the fusion partner. Additionally, the secretion capacity of *E. coli* is less effective than that of eukaryotic hosts for achieving a high yield and does not favor its manufacture on an industrial scale.

Yeast and filamentous fungi expression systems are convenient for preparing high yields of VHHs [40]. However, VHH expression in yeast depends on its secretion efficiency, and excessive mannosylation mechanisms in yeast may impair VHH antigen-binding characteristics [40–43]. *Aspergillus oryzae* (*A. oryzae*), a filamentous fungus, has been granted a Genetically Regarded as Safe (GRAS) status by the Food and Drug Administration (FDA) [44]. *A. oryzae* has a larger genome than other *Aspergillus* species due to genes encoding secretory hydrolases [45]. In contrast to *Saccharomyces cerevisiae* (*S. cerevisiae*), another GRAS organism that is used to produce commercial VHH antibodies, *A. oryzae* has a unique secretion mechanism that facilitates downstream processes and enables industrial-scale production of heterologous proteins [46]. The *A. oryzae* expression system requires less expensive media than the mammalian system, can produce complex proteins, and

is resistant to a variety of environmental conditions [47]. Human antibodies have been successfully expressed in *A. oryzae* [48].

*A. oryzae* has become essential as an expression system in consideration of the presence of auxotrophic nutrient markers such as *pyrG*, which can be restored by gene replacement, and promoter genes such as *glaA* (*glucoamylase A*) that ensure a high yield in addition to effective transformation methods [45,49,50]. Therefore, *pyrG* auxotrophic *A. oryzae* can play an important role in the industrial production of functional VHH antibodies required for many biotechnological studies and applications [47].

The objective of this study is to use the superiority of a *pyrG* auxotrophic (-) *A. oryzae* RIB40 strain in heterologous protein production to produce large amounts of recombinant functional anti-ribonuclease A (anti-RNase A) VHH, one of the nanobodies with a known structure and antigen-binding affinity [51,52]. Herein, we aim to demonstrate that *A. oryzae* is a useful platform for the large-scale production of VHHs, with a high binding affinity, for use as affinity reagents.

## 2. Materials and Methods

*E. coli* TOP10 (C404010) was purchased from Thermo Fisher Scientific (Waltham, MA, USA). *A. oryzae* strain RIB40 (42149) was purchased from ATCC (American Type Culture Collection, Manassas, VA, USA). Luria broth (LB) containing 1% tryptone, 0.5% yeast extract, and 1% NaCl supplemented with 100 μg/mL ampicillin was used as a growth medium for the bacteria. Czapek-Dox (CD) agar medium containing 0.2% $NaNO_3$, 0.1% $K_2HPO_4$, 0.05% $MgSO_4.7H_2O$, 0.05% KCl, 0.001% $FeSO_4.7H_2O$, 3% sucrose, 5% NaCl, and %2 agar (pH 5.5) was used as the minimal medium of *A. oryzae*. DPY medium containing 2% dextrin, 1% polypeptone, 0.5% yeast extract, 0.5% $KH_2PO_4$, and 0.05% $MgSO_4.7H_2O$ (pH 5.5) was used as a growth medium for *A. oryzae*. The DPY medium (2×) containing 4% dextrin, 1% polypeptone, 0.5% yeast extract, 0.5% $KH_2PO_4$, and 0.05% $MgSO_4.7H_2O$ (pH 5.5) was used for expression. Auxotrophic *A. oryzae* was grown in a DPY medium supplemented with 20 mM uridine and 0.2% uracil. The phosphate-buffered saline (PBS) buffer containing 137 mM NaCl, 2.7 mM KCl, 10 mM $Na_2HPO_4$, and 1.8 mM $KH_2PO_4$ (pH 7.4) was used for size exclusion chromatography as equilibration, wash, and elution buffers. The nucleotide sequences were synthesized by GenScript Biotech PTE. LTD. (Piscataway, NJ, USA).

The chemicals and reagents were purchased from Sigma (St. Louis, MO, USA). RNase A was purchased from Biofroxx (Einhausen, Germany, 1263MG050). The fungal cell wall lysing enzyme (Yatalase, T017) was purchased from Takara Bio Inc. (Shiga, Japan). Plasmid midiprep kits used for isolating plasmids and nickel resin (HisPur™ Ni-NTA Resin) used for purifying recombinant proteins were purchased from Thermo Fisher Scientific (Waltham, MA, USA). The restriction enzymes and the DNA ligation kit were purchased from New England Biolabs (Ipswich, MA, USA). HRP-conjugated anti-6X His tag® antibody (ab1269) was purchased from Abcam (Cambridge, UK). For Western blotting, a nitrocellulose membrane (Amersham™ Protran® Premium, 10600003) was purchased from GE Healthcare (Chicago, IL, USA) and a chemiluminescence detection kit (Western Bright™ Sirius, K-12043-D10) was purchased from Advansta (San Jose, CA, USA). Gangnam-Stain Protein Ladder (24052) was purchased from Intron Biotechnology (Seongnam, Republic of Korea) and used for sodium dodecyl sulfate polyacrylamide gel electrophoresis (SDS-PAGE) with 15% gels. Amicon® Ultra-15 centrifugal filters were purchased from Merck Millipore (Burlington, MA, USA). Ultrafiltration cassettes (Sartocon® Slice 200) were purchased from Sartorius Stedim Biotech (GÖ, Germany). PD-10 desalting column and Superdex 200 Increase 10/300 GL column were purchased from GE Healthcare (Chicago, IL, USA). A 6 L fermenter (INFORS HT, Minifors 2, Bottmingen, Switzerland) was used for fermentation studies.

### 2.1. Construction of Auxotrophic A. oryzae Strain by the Deletion of pyrG

The *pyrG* gene encodes orotidine 5′-monophosphate decarboxylase, which is required for *A. oryzae* survival, via the pyrimidine pathway. Moreover, 5-fluoroorotic acid (5-FOA)

is toxic for *A. oryzae* because it participates in the transformation of a pyrimidine into an inhibitor of the DNA precursor thymidylate, via orotidine 5′-monophosphate decarboxylase. As a result, *pyrG(-) A. oryzae* cannot survive without uridine and uracil but is resistant to 5-FOA [53]. In this study, the *pyrG* gene was chosen for rendering auxotrophy because it is a safe selective marker that can be recovered by gene replacement through transformation [54].

The genomic DNA of wild-type *A. oryzae* RIB40 was used as a template for the construction of the *pyrG* gene-deletion cassette (Figure 1). The *pyrG* gene (~1.7 kb) is located on chromosome 7. The GenBank accession number for *pyrG* is GQ496621.1. The upstream region (1.3 kb) of the *pyrG* gene was amplified using PCR primer pairs $P_{5'F}$ and $P_{5'R}$, including restriction sites *HindIII* and *SbfI*. The downstream region (1.2 kb) of the *pyrG* gene was amplified by PCR with primer pairs $P_{3'F}$ and $P_{3'R}$, including restriction sites *SbfI* and *EcoRI*. The amplified DNA fragments were digested and then ligated together. The product was then cloned into the backbone of the pUC19 vector. Sequencing was used to confirm the sequence of the deletion vector (pUC19_*pyrG*(-)).

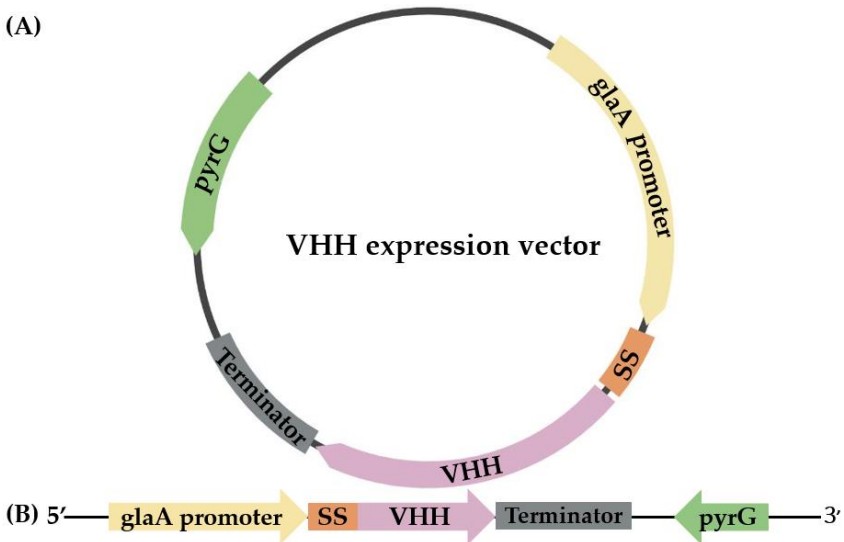

**Figure 1.** A schematic representation of circular (**A**) and linear (**B**) VHH expression vectors comprising the glucoamylase (glaA) promoter, the glucoamylase signal sequence (SS), the gene encoding anti-RNase A VHH (VHH), the terminator, and the *pyrG* gene.

The pUC19_*pyrG*(-) vector was amplified in and then isolated from *E. coli* TOP10. Following that, the deletion vector was linearized by *HindIII* and *EcoRI* digestion for *A. oryzae* transformation.

The transformation was performed according to Sakai [55]. In brief, *A. oryzae* RIB40 was grown overnight in DPY medium at 30 °C and 180 rpm. After incubation, mycelia were lysed using a lysis solution (50 mM malate buffer, 0.6 M $(NH_4)SO_4$, and 1% yatalase; pH 5.5) at 30 °C and 80 rpm for 4 h. The spheroplasts were grown overnight in DPY medium, harvested, and washed with a wash solution of 1.2 M sorbitol and 50 mM $CaCl_2$. The obtained spheroplasts and linearized pUC_19-*pyrG*(-) vector were mixed in the presence of PEG4000, and the mixture was then spread on CD agar plates. The control group was treated without the addition of insert DNA. Incubation was performed at 30 °C for 5–7 days.

The *pyrG* gene was deleted according to Ling [54] with the following modifications: After transformation, all conidia were immediately spread on the non-selective regeneration media containing 20 mM uridine and 0.2% uracil but not 5-FOA to ensure the survival of transformed colonies among the untransformed colonies. Following that, all colonies (a mix of transformed and non-transformed colonies) were harvested, diluted ($4 \times 10^6$ cells), and pre-cultured in DPY medium with 20 mM uridine and 0.2% uracil to eliminate non-

transformed colonies and ensure the growth of transformed colonies. After overnight incubation at 30 °C and 180 rpm, the pre-culture was spread on selective CD agar plates containing 20 mM uridine, 0.2% uracil, and 1.6 mg/mL 5-FOA. After 5–7 days of incubation, the 5-FOA-resistant individual colonies were collected and plated again on selective CD agar plates containing uridine, uracil, and 5-FOA to check the stability of auxotrophy. For the control group, the same 5-FOA-resistant colonies were plated out on non-selective CD agar plates and cultured in a non-selective DPY medium without uracil or uridine. To confirm the deletion of the *pyrG* gene, genomic DNA from 5-FOA-resistant individual colonies was isolated and tested by qPCR analysis, performed using primer pairs designed according to the upstream and downstream regions of the *pyrG* gene in the RIB40 genome.

## 2.2. Cloning and Expression of VHH in pyrG(-) A. oryzae

In this study, a VHH antibody that selectively binds RNase A was chosen to be expressed in *pyrG(-) A. oryzae*. The amino acid sequence of the selected anti-RNase A VHH was obtained from the NCBI database (accession number 2P49_B). The DNA sequence encoding the anti-RNase A VHH was codon-optimized, and an 8xHis-tag was inserted at the C terminus. The codon-optimized sequence was synthesized at GenScript Biotech PTE. LTD. (Piscataway, NJ, USA) and inserted into an expression vector harboring the glucoamylase promoter (Figure 1).

The expression vector of anti-RNase A VHH was digested by *HindIII* and *EcoRI*, and the obtained linearized expression vector was transformed into *pyrG(-) A. oryzae* according to Sakai [55]. A control gene cassette containing only the *pyrG* gene (without any insert gene) was also prepared and transformed (mock transfection control) into *pyrG(-) A. oryzae*. After transformation, the colonies were transferred to 15 mL of DPY medium and incubated overnight at 30 °C and 180 rpm. Subsequently, each culture was diluted in 75 mL of 2× DPY at a ratio of 1:10. Incubation was performed at 30 °C and 180 rpm for 7 days. After incubation, samples were taken from the supernatant of the expression cultures and analyzed using SDS-PAGE. Anti-RNase A VHH, which was expressed in *pyrG* auxotrophic *A. oryzae*, was referred to as *Asp.* VHH in this study.

For fermentation, the colony expressing *Asp.* VHH was grown overnight in a 400 mL medium containing 20 g/L dextrin, 10 g/L peptone, 5 g/L yeast extract, 1.5 g/L $KH_2PO_4$, and 0.15 g/L $MgSO_4.7H_2O$ at 30 °C and 180 rpm. After incubation, the culture was diluted in 4 L of growth medium containing 30 g/L dextrin, 7.5 g/L $(NH_4)_2SO_4$, 3 g/L yeast extract, 1.5 g/L $KH_2PO_4$, 1 g/L peptone, 1 g/L $MgSO_4.7H_2O$, 1 g/L NaCl, 0.1 g/L $CaCl_2.2H_2O$, 0.5 mL/L trace element solution, and 1 mL/L antifoam at a ratio of 1:10. The trace element solution consisted of 10.75 g/L $ZnSO_4.7H_2O$, 1.9 g/L $CuSO_4.5H_2O$, 0.38 g/L $NiCl_2.6H_2O$, and 10.4 g/L $FeSO_4.7H_2O$. During fermentation, the temperature, pH, and dissolved oxygen content of the culture were kept constant at $30 \pm 0.5$ °C, 5.5, and $20 \pm 2\%$, respectively. Aeration was accomplished using air flow at a rate of 1 L/min. The agitator ran between 400 and 750 rpm. The fermentation process was terminated after 120 h.

## 2.3. Purification of the VHH Expressed in A. oryzae

The medium containing the proteins released by *A. oryzae* was obtained by filtering the culture of *A. oryzae* expressing *Asp.* VHH via Whatman filter paper, which removes fungal cells. The culture medium thus obtained was concentrated using an ultrafiltration membrane (Sartocon® Slice 200) with a cut-off of 5000 molecular weight cut-off (MWCO) and purified using metal affinity chromatography (IMAC) under native conditions, taking advantage of the 8xHis-tag at the C-terminus of the recombinant *Asp.* VHH. The concentrated sample was incubated for 2 h at 4 °C with HisPur™ Ni-NTA resin. Following incubation, the resin–sample combination was packed into a nickel–nitrilotriacetic acid (Ni–NTA) affinity column and rinsed with a wash buffer containing 50 mM $NaH_2PO_4$ and 500 mM NaCl (pH 7.4). An elution solution comprising 50 mM $NaH_2PO_4$, 300 mM NaCl, and 200 mM imidazole (pH 7.4) was used to elute the bound 8xHis-tagged protein. To remove imidazole, the elution fractions were desalted on a PD10 desalting column

according to the manufacturer's instructions, using a PBS buffer (pH 7.4). The obtained samples from the purification processes were examined using 15% SDS-PAGE to ensure that they were sufficiently pure.

To evaluate the composition and purity of the eluted *Asp*. VHH, the fractions containing *Asp*. VHH was applied to a Superdex 200 Increase 10/300 GL size exclusion chromatography column that was equilibrated using a running buffer of PBS (pH 7.4), which was conducted in accordance with the manufacturer's instructions. *Asp*. VHH was eluted using this buffer at a linear flow rate of 0.5 mL/min. The eluted fractions were concentrated using an Amicon® Ultra-15 centrifugal filter, and the purity of *Asp*. VHH was assessed using SDS-PAGE and Western blot.

Densitometric analysis of total protein production was performed using the ImageJ 1.53t tool (Version 1.53t, National Institutes of Health, Bethesda, MD, USA). Bradford assay was used to measure the concentration of total protein and pure *Asp*. VHH at 595 nm using bovine serum albumin as a protein standard.

### 2.4. Western Blot Analysis of Asp. VHH

HRP Anti-6X His tag® antibody (1:5000) was used in a Western blot assay. Samples were separated on a 15% SDS-PAGE gel and transferred to a nitrocellulose membrane using a transfer solution comprising 24 mM Trizma Base, 192 mM glycine, and 20% methanol. Blocking was performed using a wash solution (10 mM Tris-HCl (pH 7.4), 0.9% NaCl, and 0.2% Tween) supplemented with 5% nonfat milk powder overnight at 4 °C. The membrane was washed three times before being treated with HRP-conjugated anti-6X His tag® antibody. The detection was carried out using a chemiluminescence Western blotting detection kit according to the manufacturer's instructions.

### 2.5. Pull down Assay

RNase A was prepared at a concentration of 1 mg/L in PBS buffer (pH 7.4). For the pull-down assay, 50 μL of HisPur™ Ni-NTA resin was pre-washed with a wash solution containing 50 mM $NaH_2PO_4$ and 300 mM NaCl (pH 7.4), combined with 100 μL of *Asp*. VHH (1.5 mg/L in PBS buffer; pH 7.4), and then incubated at 4 °C for 1 h (experimental resin). In the absence of *Asp*. VHH, 50 L of pre-washed HisPur™ Ni-NTA resin was incubated with only 100 μL of PBS buffer (pH 7.4) as a negative-control resin. Following incubation, the resins were washed three times, for 2 min each time, with the wash solution at $700 \times g$ at 4 °C. Then, 150 μL of RNase A solution was added to the resins (experimental and negative control) and the resulting mixture was incubated at 37 °C for 1 h. After the three rounds of washing, the bound proteins were released using an elution solution containing 50 mM $NaH_2PO_4$, 300 mM NaCl, and 300 mM imidazole (pH 7.4). The collected samples were analyzed using SDS-PAGE gel.

### 2.6. Size Exclusion Assay for Determining the Specificity of Asp. VHH against RNase A

To determine the specificity of *Asp*. VHH against RNase A, as a first step, pure *Asp*. VHH was applied to a Superdex 200 Increase 10/300 GL size exclusion chromatography column. In a subsequent step, RNase A was prepared in PBS buffer (pH 7.4) and administered separately to the Superdex 200 Increase 10/300 GL column. Finally, *Asp*. VHH and RNase A were mixed in a ratio of 1:5 and incubated at 37 °C for 30 min. Following incubation, the mixture of *Asp*. VHH and RNase A were applied to the column. At room temperature, all size exclusion chromatography assays were carried out at a linear flow rate of 0.5 mL/min. The running buffer was PBS buffer, pH 7.4.

### 2.7. Surface Plasmon Resonance (SPR) Analysis for Measuring the Binding Affinity of Asp. VHH and RNase A

Surface plasmon resonance (SPR) was performed using a BIAcore T200 system (Cytiva, Sweden) with the commercial Series S sensor chip CM5. For kinetic measurements, in-house expressed and purified *Asp*. VHH was captured on flow cell 4 with non-specific

interactions at a flow rate of 10 µL/min for 60 s. To evaluate binding kinetics and avoid bulk effects, the multi-cycle kinetics method was used with RNase A at the following dilutions: 0, 2, 10, and 50 nM. For all concentrations, RNase A was injected through flow cells 3 and 4 with an association time of 75 s and a dissociation time of 225 s at a flow rate of 30 µL/min. HBS-EP+ (10 mM HEPES with a pH of 7.4, 150 mM NaCl, 3 mM EDTA, 0.005% surfactant P20) was used as the running buffer, and all dilutions were prepared with this buffer. Binding reactions were performed at 25 °C, and the samples were stored at 12 °C. Binding kinetics were calculated using Biacore T200 Evaluation Software (version 3.2.1, Cytiva) with the 1:1 binding model.

## 3. Results

### 3.1. The pyrG Auxotrophic A. oryzae

The *pyrG* deletion approach is based on homologous recombination between the 5′ and 3′ flanking regions of the deletion vector and the exact same flanking areas of the *pyrG* gene in the *A. oryzae* wild-type genome [54].

The *pyrG* deletion vector was created by combining the upstream and downstream flanking regions of the *pyrG* gene, as shown in Figure 2. The 5′ upstream flanking area was amplified using primers including the restriction enzyme sites *HindIII* and *SbfI*, and the 3′ downstream flanking region was amplified using primers containing the restriction enzyme sites *SbfI* and *EcoRI*. The amplified areas were assembled into a pUC19 vector backbone without a *pyrG* open reading frame (pUC19_*pyrG*(-)) following restriction enzyme digestion and subsequent ligation.

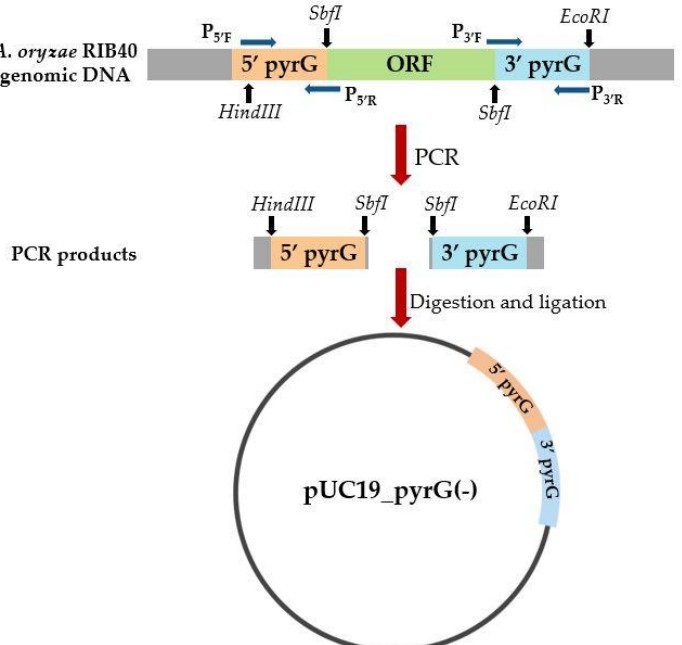

**Figure 2.** The construction of the *pyrG* deletion vector (pUC19_*pyrG*(-)) is schematically depicted. The *pyrG* deletion vector was created by combining the 1.3 kb upstream fragment (5′ *pyrG*) and 1.2 kb downstream fragment (3′ *pyrG*) of *A. oryzae* RIB40 genomic DNA. PCR was used to amplify the 5′ and 3′ *pyrG* sequences without the *pyrG* open reading frame (ORF). P$_{5′F}$–P$_{5′R}$ and P$_{3′F}$–P$_{3′R}$ are primer pairs for 5′ *pyrG* and 3′ *pyrG*, respectively. To obtain pUC19_*pyrG*(-), PCR products were digested with *HindIII*, *SbfI*, and *EcoRI* restriction enzymes and ligated into the pUC19 vector backbone.

The *pyrG* auxotrophic *A. oryzae* strain was generated by transforming *A. oryzae* wild-type RIB40 with the pUC19_*pyrG*(-) deletion vector [54]. As a result of having an inactive *pyrG* in the pyrimidine pathway, *pyrG(-)* colonies require uridine and uracil to survive and develop resistance to 5-FOA (Figure 3). Wild-type colonies, however, do not require

uridine or uracil to survive and cannot live in the presence of 5-FOA since their pyrimidine pathway is active (Figure 3).

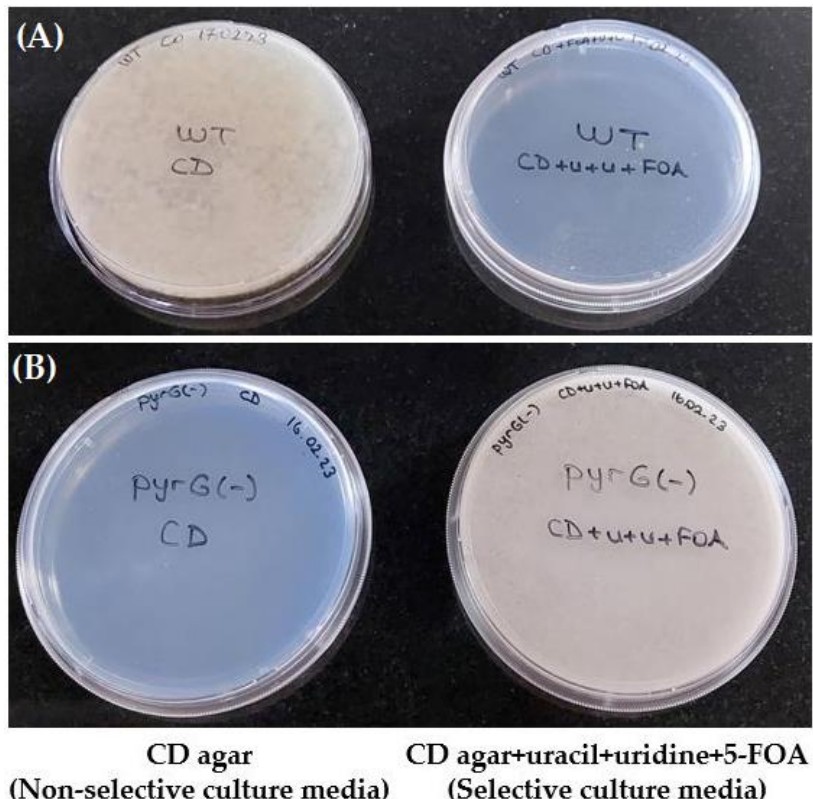

**CD agar**
**(Non-selective culture media)**

**CD agar+uracil+uridine+5-FOA**
**(Selective culture media)**

**Figure 3.** An illustration of the growth potential of *pyrG(-)* and wild-type (WT) *A. oryzae* on selective and non-selective culture media. CD agar culture medium plates are non-selective. CD agar plates of culture medium mixed with uracil, uridine, and 5-FOA are selective. (**A**) Wild-type *A. oryzae* can grow on non-selective media (left) but not on selective media in the presence of 5-FOA (right). (**B**) *pyrG(-)* *A. oryzae* cannot grow on a non-selective media lacking uracil and uridine (left) but can grow on selective media in the presence of uracil, uridine, and 5-FOA (right). (U+u+FOA: uracil, uridine and 5-fluoroorotic acid).

Three consecutive rounds of re-streaking *pyrG(-)* colonies on selective CD agar demonstrate the stability of the *pyrG* deletion. Both selective and non-selective DPY media were used to examine the growth potential of 5-FOA-resistant single colonies. Cultures were started at an $OD_{600}$ of 0.4, and after overnight incubation at 30 °C and 180 rpm, the $OD_{600}$ of *pyrG(-)* colonies in the selective DPY medium with uracil and uridine was measured at 4.34, whereas no significant changes were observed in the $OD_{600}$ of *pyrG(-)* colonies in the DPY medium without uracil and uridine. qPCR was used to confirm the deletion of the *pyrG*-encoding gene in the 5-FOA-resistant colonies. Furthermore, the 5-FOA-resistant colonies were tested using qPCR to determine whether the *pyrG*-encoding gene had been eliminated.

### 3.2. The Expression of VHH in pyrG(-) A. oryzae

The *pyrG* gene locus was inserted by gene replacement into the *pyrG(-)* *A. oryzae* genome via the expression vector encoding *pyrG* and *Asp*. VHH. Protein expression tests were performed on several transformed colonies in a small flask volume after transformation. SDS-PAGE examination of medium samples after 7 days of incubation revealed that *Asp*. VHH was successfully expressed in multiple *A. oryzae* colonies (Figure 4A). Figure 4B displays both the SDS-PAGE profile of a selected recombinant colony expressing *Asp*. VHH and mock transfection control.

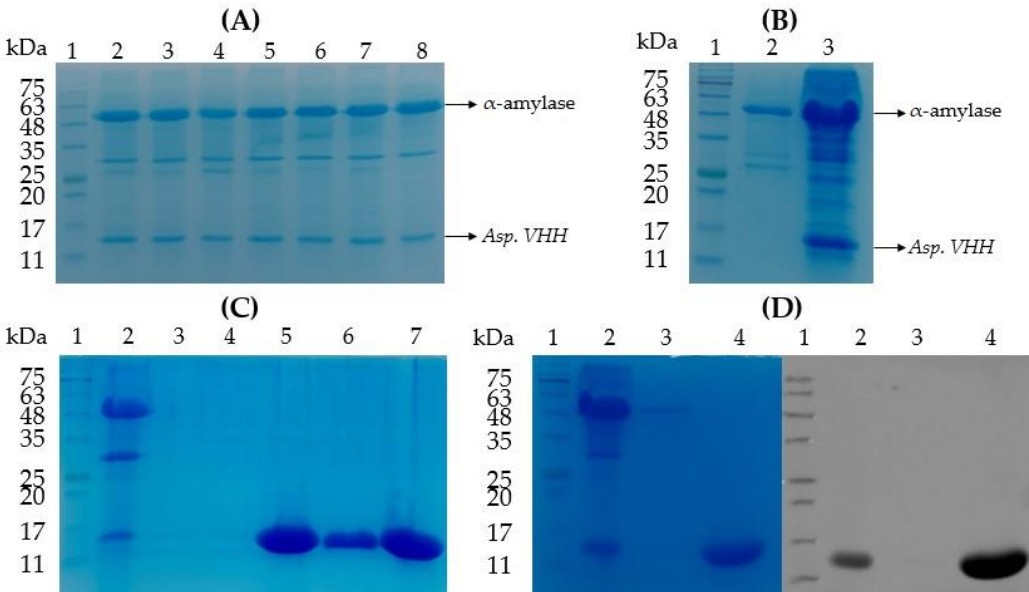

**Figure 4.** Sodium dodecyl sulfate polyacrylamide gel electrophoresis (SDS-PAGE) and Western blot studies of 8xHis-tagged *Asp*. VHH (~13 kDa) expressed in *pyrG(-) A. oryzae*. (**A**) SDS-PAGE profile of the small-scale protein expression of multiple transformed colonies after *A. oryzae* transformation. Lane 1: Gangnam-Stain Protein Ladder. Lanes 2–8: Culture medium samples from transformed colonies expressing *Asp.* VHH. (**B**) SDS-PAGE profile of the mock transfection control and a recombinant colony expressing *Asp*. VHH. Lane 1: Gangnam-Stain Protein Ladder. Lane 2: Culture medium of the mock transfection control. Lane 3: Culture medium of the recombinant colony expressing *Asp*. VHH. (**C**) SDS-PAGE analysis of purified *Asp*. VHH from large-scale production in a 6-liter fermenter by IMAC. Lane 1: Gangnam-Stain Protein Ladder. Lane 2: Concentrated culture medium sample of the recombinant colony expressing *Asp*. VHH. Lanes 3 and 4: The samples obtained from the IMAC wash steps. Lanes 5–7: The eluted sample containing *Asp*. VHH collected from the IMAC column. (**D**) Verification of purified *Asp*. VHH by SDS-PAGE (left) and Western blot (right) analyses, where for both the gel and blot, lane 1 is the Gangnam-Stain Protein Ladder, lane 2 is the culture medium of the recombinant colony expressing *Asp*. VHH, lane 3 is the sample obtained from IMAC wash step, lane 4 is the eluted sample containing *Asp*. VHH collected from the IMAC column.

Recombinant *Asp*. VHH was successfully expressed in *A. oryzae* and secreted into the culture medium, as shown in Figure 4A,B. Its molecular weight is approximately 13 kDa. The α-amylase, approximately 52 kDa, is a natural product secreted into the culture medium by *A. oryzae.* Once the small-scale expression was successfully completed, the 6 L fermenter was used to obtain a larger amount of *Asp*. VHH, and IMAC was used to purify the expressed *Asp*. VHH (Figure 4C).

### 3.3. The Purification of Asp. VHH

To purify the recombinant *Asp*. VHH secreted by *A. oryzae* into the culture medium, metal affinity chromatography was performed. The culture media was filtered to separate the released proteins from the fungal cell debris. To separate the expressed 8xHis-tagged *Asp*. VHH from other *A. oryzae* host proteins, a metal-affinity chromatography column was used. To increase the amount of recombinant *Asp*. VHH attached to the resin, the culture sample and resin were incubated together. As expected, SDS-PAGE analysis indicated the existence of a protein band with a molecular weight of approximately 13 kDa, which is *Asp*. VHH (Figure 4C,D). SDS-PAGE (Figure 4C) and Western blot (Figure 4D) were used to determine the purity of the *Asp*. VHH. The purified recombinant *Asp*. VHH yielded 44 mg/L in shake flask expression and 1.4 g/L in fermentation expression.

Size exclusion chromatography was used to achieve a high level of purification and to assess the monomeric structure and/or degradation state of *Asp*. VHH. According to the

chromatogram (Figure 5), the recombinant *Asp*. VHH was eluted in a single peak, reflecting its monomeric behavior.

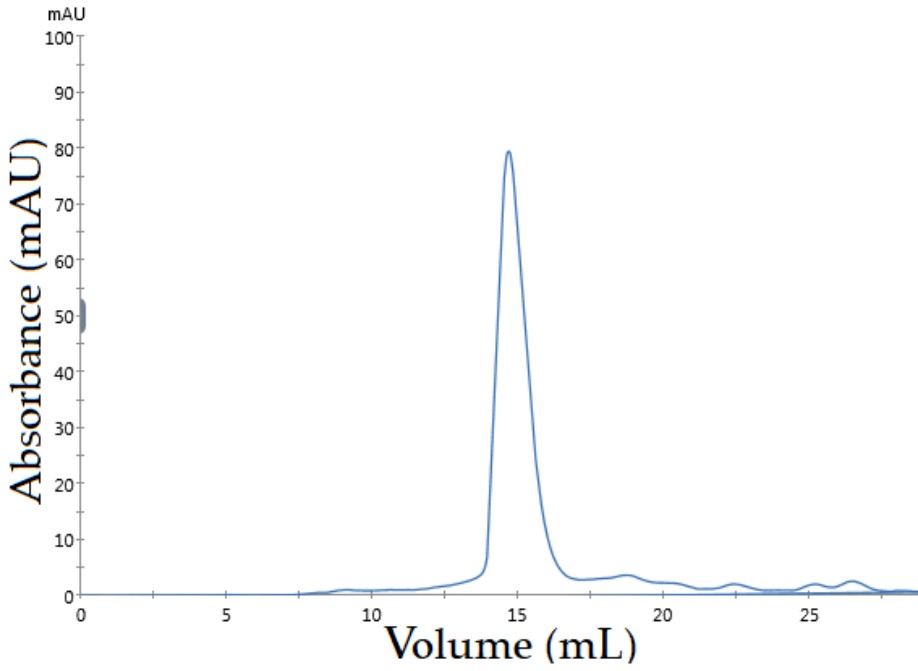

**Figure 5.** Chromatogram of size exclusion chromatography using a Superdex 200 Increase 10/300 GL column. The peak represents pure *Asp*. VHH.

*3.4. Determining the Specificity of Asp. VHH against RNase A*

To determine the specificity of *Asp*. VHH against RNase A as its target, a pull-down assay was conducted under native conditions. In this assay, *Asp*. VHH but not RNase A can bind to resin, via its C-terminus 8xHis-tag. Figure 6 illustrates the samples obtained from the various stages of the pull-down assay on the SDS-PAGE gel. *Asp*. VHH was specifically attached to the experimental resin, and RNase A then interacts with *Asp*. VHH. As predicted, *Asp*. VHH and RNase A co-eluted from the experimental resin (Figure 6, lane 8). In Figure 6, lane 8, the upper protein band corresponds to RNase A and the lower protein band to *Asp*. VHH, as expected. In the negative-control resin, RNase A was eliminated from the resin during the flow-through and washing stages (Figure 6, lanes 5 and 7) and did not interact with the resin in the absence of *Asp*. VHH. The pull-down assay results demonstrate that recombinant *Asp*. VHH expressed in *A. oryzae* was specifically bound to RNase A.

In addition to the pull-down assay, size-exclusion chromatography confirmed the specificity of *Asp*. VHH for RNase A. The purified *Asp*. VHH and RNase A were individually applied to the Superdex 200 Increase 10/300 GL column to create protein-specific chromatograms. Figure 7A is the chromatogram for *Asp*. VHH, and Figure 7B is the chromatogram for RNase A. To investigate the interaction of VHH and RNase A, they were incubated together. Following incubation, a chromatogram of the complex of *Asp*. VHH and RNase A was obtained (Figure 7C). These results are consistent with expectations based on the difference in the molecular weights of *Asp*. VHH (~13 kDa) and RNase A (~13.7 kDa).

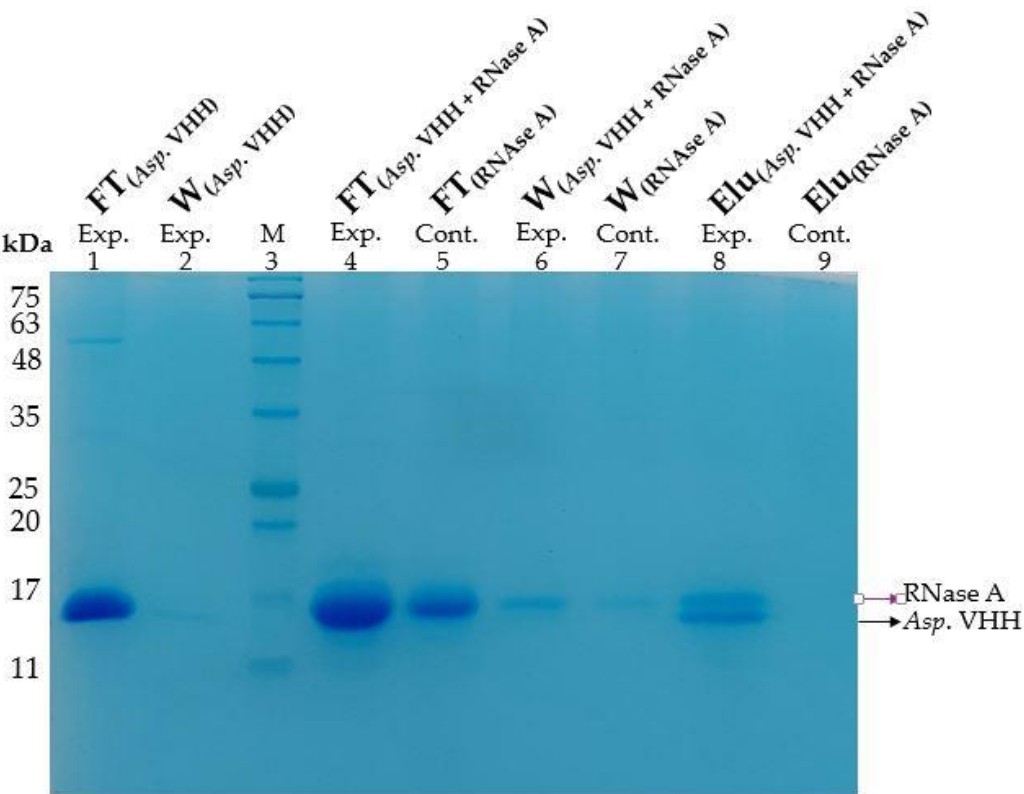

**Figure 6.** Pull-down assay to investigate the specificity of *Asp*. VHH against RNase A. The control resin (Cont.) was incubated with only RNase A. The experimental resin (Exp.) was co-incubated with *Asp*. VHH and RNase A. Lane 1: Flow-through sample containing only *Asp*. VHH (experimental resin). Lane 2: Wash sample obtained from the experimental resin after incubation with *Asp*. VHH. Lane 3: Gangnam-Stain Protein Ladder. Lane 4: Flow-through sample collected after co-incubation of *Asp*. VHH and RNase A (experimental resin). Lane 5: Flow-through sample containing only RNase A (control resin). Lane 6: Wash sample obtained from the experimental resin after RNase A incubation. Lane 7: Wash sample obtained from the control resin after RNase A incubation. Lane 8: Elution sample collected from the experimental resin containing RNase A and *Asp*. VHH. Lane 9: Elution sample collected from the control resin. Exp: experimental resin, Cont: control resin, FT: Flow-through sample, W: wash sample, and Elu: elution sample.

*Asp*. VHH and RNase A combined to create a complex, as seen in Figure 7C, with a molecular weight greater than that of RNase A alone. The peak corresponding to *Asp*. VHH, which can be seen in Figure 7A, was not obtained in Figure 7C because *Asp*. VHH molecules combined with RNase A. The results of the size-exclusion assay confirm that *Asp*. VHH expressed in *A. oryzae* successfully forms a complex with its target molecule, and they are supported by the results of the pull-down assay.

### 3.5. Determining the Binding Affinity of Asp. VHH to RNase A

The binding kinetics of the *Asp*. VHH protein in targeting RNase A was determined using surface plasmon resonance (SPR). The *Asp*. VHH protein was captured on a CM5 sensor chip before being injected with a concentration series of RNase A. The sensorgrams (Figure 8) were fitted to a 1:1 binding model and the binding affinity (Kd) of *Asp*. VHH was calculated to be 1.91 nM.

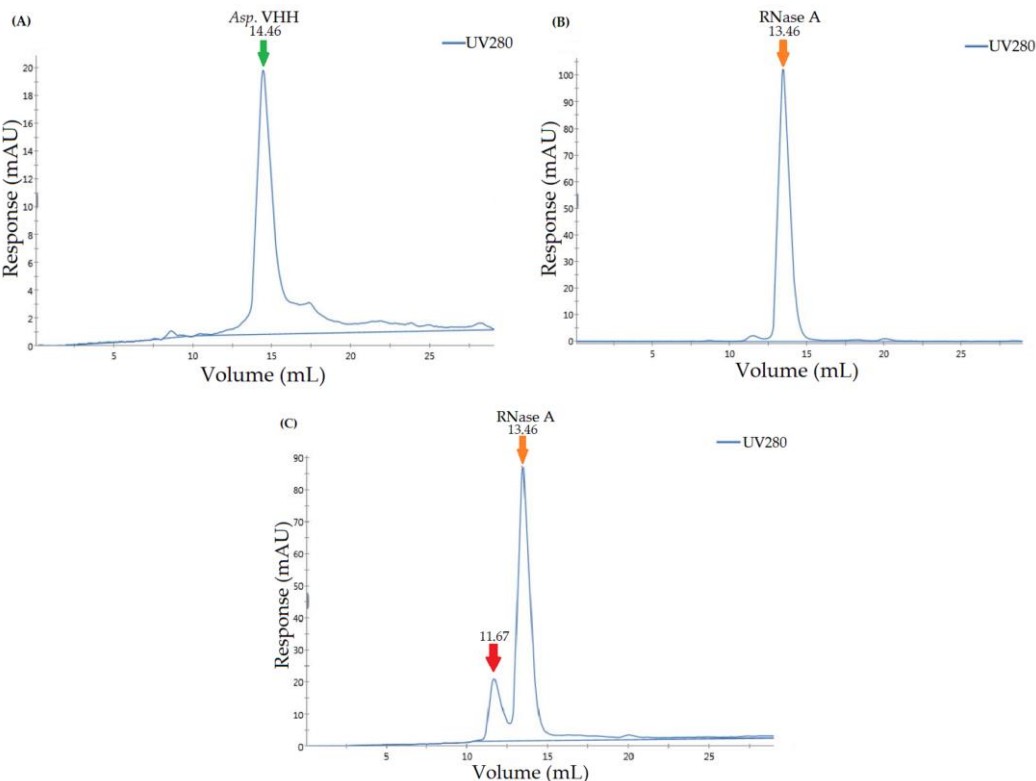

**Figure 7.** Size-exclusion chromatographic profiles on the Superdex 200 Increase 10/300 GL column of *Asp*. VHH, RNase A, and the mixture of *Asp*. VHH and RNase A. (**A**) *Asp*. VHH was applied to the size exclusion column; the green arrow denotes the peak of *Asp*. VHH. (**B**) RNase A was applied to the size exclusion column; the orange arrow denotes the peak of RNase A. (**C**) After *Asp*. VHH and RNase A were incubated together, the mixture was applied to the size exclusion column. The orange arrow represents the peak of RNase A, while the red arrow represents the complex of *Asp*. VHH and RNase A.

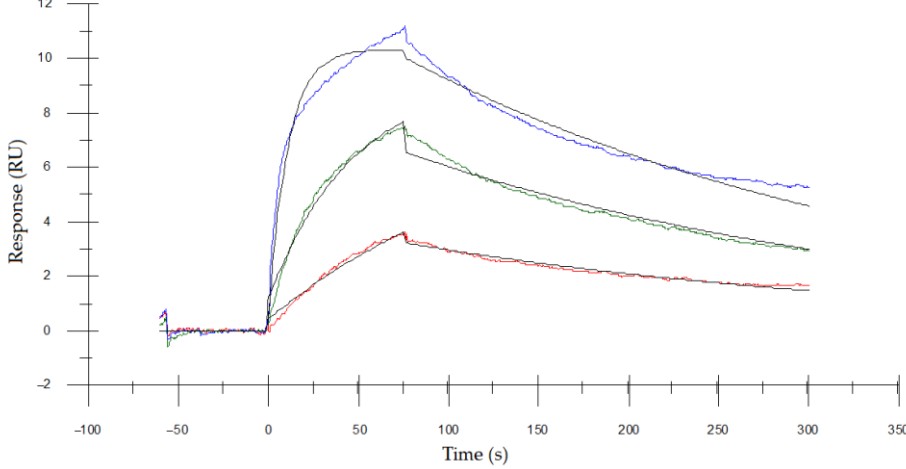

**Figure 8.** Sensorgrams display measured curves, whereas black lines depict fitted curves. The red curve represents the lowest RNase A concentration (2 nM), the green line indicates mid-concentration of RNase A (10 nM), whereas the blue curve represents the highest RNase A concentration (50 nM). A blank cycle was subtracted from each curve, and the binding activity of RNase A was measured. *Asp*. VHH was captured on the surface of the flow cell. The kinetics demonstrates an interaction between increasing concentrations of RNase A and *Asp*. VHH.

## 4. Discussion

In our study, we obtained 44 mg/L of pure *Asp*. VHH in a shake flask and 1.4 g/L in a fermenter by utilizing the capabilities *A. oryzae* in secreting the expressed protein into the medium, from which it was efficiently purified under native conditions without the requirement for lysis or refolding.

The regulation of *pyrG* in the pyrimidine pathway and its impact on the amount of VHH antibodies were also examined in *E. coli* [56]. Landerg et al. demonstrated that using a CRISPR single-guide RNA (sgRNA) to repress *pyrG* in *E. coli* increases the VHH antibody production yield by 2.6-fold compared with an *E. coli* strain lacking sgRNA by maintaining ribosome content and decreasing cell growth [56]. It should be highlighted that the indefinite reusability of the *pyrG* selectable marker via gene replacements enables a safe and sustainable practice for *A. oryzae*, which is a GRAS organism used for industrial applications [54,57].

In our investigation, we used *A. oryzae* as an expression platform for functional *Asp.* VHH since it is capable of expressing heterologous proteins at the industrial level and supports the post-translational modifications necessary for proper folding of one of VHH antibodies carrying a disulfide bond. The yield of VHH antibodies in *E. coli* is quite variable [34] and depends on the type of VHH, fusion tag used, or production strategies, resulting in complex downstream processes (Table S1) [37,45,58–63]. In one of these studies, Salema et al. discovered that the expression level of VHH antibodies was dramatically altered in *E. coli*. They created a recombinant fusion VHH with maltose-binding protein (MBP) to achieve a high expression level (12 mg/L) and discovered that MBP-tagged VHH exhibited lower affinity than non-fusion VHH antibodies [38]. Studies [37,63] that sought to obtain biologically active VHH antibodies in significant quantities have been taken into consideration in *E. coli* (Table S1). Although *E. coli* is not recommended for industrial-scale production, *A. oryzae* has been used as a GRAS organism in industrial fermentation technology for many years [41,44,64].

In heterologous protein production, post-translational modification is an important stage that should be examined to guarantee that the produced protein retains its functionality and that its activity is not affected by glycosylation. In this regard, eukaryotic hosts outperform bacterial expression systems. However, due to the time-consuming and complex production processes, it is not feasible to use mammalian expression on a large scale [34]. Yeast expression has been successfully used to generate functional VHH with disulfide bonds (Table S1). In a shake flask, Frenken et al. [65] obtained a yield of more than 100 mg/L of VHH in *S. cerevisiae*. In addition to *S. cerevisiae*, *P. pastoris* has also been used to express VHH antibodies [42,43,66]. Rahbarizadeh et al. [67] showed, for the first time, that *P. pastoris* may be used for the large-scale (10–15 mg/L) manufacture of anti-MUC1 VHH molecules against breast cancer [67]. In contrast to the glycosylation process used by *S. cerevisiae*, which has a high mannose concentration, filamentous fungus can produce proteins with short oligo-mannose glycans [68]. *A. oryzae* also grows on a low-cost sugar-based minimum medium and does not require inducers such as methanol for *P. pastoris*, making it a cost-effective platform. The key consideration for using *A. oryzae* as a host is that the VHH expression level in yeast varies with secretion efficiency, whereas *A. oryzae* has a high secretion capacity [41,64].

Many factors influence the yield of an antibody fragment during heterologous expression, including the coding sequence of a recombinant protein, the design of the expression plasmid, folding mechanisms, rate of degradation, and resistance to environmental changes [33]. Joosten [69] et al. were the first to report the expression of non-fused VHH with the ability to bind azo dye in *Aspergillus awamori (A. awamori)*. They obtained a limited amount of secreted VHHs (7.5 mg/L) and discovered that the final yield decreased because of the 80% degradation and VHH adhesion to fungal cell walls [69]. In another investigation, they created peroxidase gene-encoded bifunctional anti-azo dye VHHs. They were successful in reducing proteolytic degradation. However, an important portion of the produced bifunctional VHHs remained intracellular in *A. awamori* due to fungal secretion

issues, which may explain the misfolding [70]. In our study, high-purity, monomeric *Asp*. VHH was expressed in *A. oryzae*. According to the findings of the size exclusion and SPR analyses conducted for this work, *Asp*. VHH was generated in the correctly folded form.

In the heterologous expression of VHHs in *A. oryzae*, the promoter used influences the level of expression. Okazaki et al. [71] demonstrated for the first time in the *niaD* mutant of *A. oryzae* the production of an EGFR-specific VHH nanobody coupled with a 28-amino acid fusion partner under the control of the manganese superoxide dismutase gene promoter [71]. Hisada et al. studied the influence of various highly secretory promoter proteins (taka-amylase A, glucoamylase, endoglunase A and B, etc.) on anti-human chorionic gonadotropin VHH expression in *A. oryzae*. They discovered that the glucoamylase produced the maximum amount of anti-human chorionic gonadotropin VHHs, and they obtained glycosylated VHH that was equivalent to expressed VHH in *E. coli* [72]. On the preferred expression platform in our study, *Asp*. VHH was expressed under the control of the glucoamylase promoter in the *pyrG(-) A. oryzae*.

The *Asp*. VHH generated in this study is a structurally recognized VHH antibody that has previously been expressed in *E. coli* [51]. We chose this VHH to test the ability of our *A. oryzae* expression system to prepare large quantities of high-purity nanobodies, as well as to assess the functionality of the recombinant VHH against its target, RNase A. The *E. coli* and *A. oryzae* expression systems were compared on the basis of the equilibrium dissociation constants of the anti-RNase VHH they generated.

In a study by Decanniere et al. [51], the equilibrium dissociation constant of anti-RNase A VHH expressed in *E. coli* was determined by SPR as Kd of 35 nM and then measured as Kd of 23 nM using the yeast surface display titration method [52]. The binding affinity of the *Asp*. VHH expressed in *pyrG* auxotroph *A. oryzae* (1.9 nM) is 18.3 times that of the VHH expressed in *E. coli* (35 nM). Because *Asp*. VHH lacks N-glycosylation sites, and the higher binding affinity of *Asp*. VHH than VHH expressed in *E. coli* may be due to variations in O-type glycosylation, an interesting possibility to address in future research.

A limitation of this study is that only one type of VHH antibody is investigated. It should be noted that the final yield for different VHHs may differ. This proof-of-principle study, however, is anticipated to influence future research on various VHH types and may offer an effective expression system for VHHs whose production in *E. coli* is problematic, such as those with a high cysteine content or an improperly folded structure.

## 5. Conclusions

VHHs are ideal and versatile affinity agents compared with mAbs because of their high affinity, stable and soluble behavior, simple structure, ease of handling, lack of an Fc component that triggers immunogenic responses, small size, and ability to reach hidden targets, as well as being able to excellently adapt to many conditions. VHHs have great potential to overcome the shortcomings of mAbs used in imaging, diagnostic, therapeutic, and biotechnological applications [12–14,73]. This brings to the forefront the establishment of a practical and economical alternative platform that will lead to the production of large quantities of functional VHHs to address the need in a wide variety of applications. Our findings indicate that the *pyrG* auxotrophic *A. oryzae* is a promising and practical alternative biotechnological platform for the production of a large number of functional VHHs with high binding activity due to its robust secretion ability, simplified product recovery via secretion products into the medium, organism safety (GRAS status), extremely efficient *pyrG* marker recycling system, and straightforward downstream processes. Additionally, this alternative expression method appears to be a good candidate for the industrial-scale production of diverse affinity reagents.

In this study, we established *pyrG* auxotrophy in *A. oryzae* through targeted gene replacement rather than chemical or random UV mutagenesis because it ensures stable and efficient transformation [57] and can serve as a reusable and safe positive selection platform for potential affinity reagents. This study has the potential to pave the way for

future research to address the demand for the efficient and cost-effective production of newly discovered affinity reagent VHHs with high binding affinity.

**Supplementary Materials:** The following supporting information can be downloaded at: https://www.mdpi.com/article/10.3390/cimb45060304/s1, Table S1: Comparison of heterologous expression systems in the field of VHH production. References [36–38,40,42,43,65–67,69–72] were in supplementary materials.

**Author Contributions:** Conceptualization, S.U.; methodology, S.U. and E.K.; validation, E.K. and A.E.E.; investigation, E.K. and A.E.E.; resources, S.U., E.K. and A.E.E.; writing—original draft preparation, E.K., L.M.A. and A.E.E.; writing—review and editing, E.K., L.M.A. and S.U.; visualization, E.K.; supervision, S.U.; project administration, E.K.; funding acquisition, S.U. All authors have read and agreed to the published version of the manuscript.

**Funding:** This study was supported by the Scientific and Technological Research Council of Turkey (TUBITAK), project number 20AG044. The APC was funded by Bezmialem Vakif University.

**Institutional Review Board Statement:** Not applicable.

**Informed Consent Statement:** Not applicable.

**Data Availability Statement:** Data sharing not applicable.

**Conflicts of Interest:** The authors declare no conflict of interest.

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
