# Peer review of "Large-Scale Production of Anti-RNase A VHH Expressed in pyrG Auxotrophic Aspergillus oryzae"

_cimb, doi:10.3390/cimb45060304_

Round 1

Reviewer 1 Report (Previous Reviewer 2)

Comments and Suggestions for Authors

The manuscript has improved significantly. The authors addressed all the issues I raised. They have added more technical details and improved the quality of the figures (figure 7 still needs improvement for the legend on the x-axis). The discussion addresses many of the issues I raised.

Author Response

Reviewer 2 Report (New Reviewer)

Comments and Suggestions for Authors

The authors of the manuscript performed a large scale production of anti-RNAse A VHH antibody using pyrG auxotrophic A. oryzae. The main issue is the scientific novelty of the performed study. Antibody expression in pyrG auxotrophic A. oryzae was reported earlier. It also concerns the method of pyrG (-) strain obtaining. As noted by the authors themselves, the disadvantage of this study is that only one type of VHH antibody is investigated. I suggest that the study should be expanded to some other VHHs.

Some minor comments:

1.      In the first paragraph of the introduction there is some confusion in the designation of antibody fragments. VH is commonly referred to as the heavy chain variable domain, but not the heavy chain. It also concerns VL.

2.      The target protein is about 13 kDa, so it is not clear which highly expressed protein represents 50 kDa band in Fig 4. The target protein should be indicated by the arrow in fig4.

Comments on the Quality of English Language

Minor editing of English language required

Round 2

Reviewer 2 Report (New Reviewer)

Comments and Suggestions for Authors

I would like to thank the Authors for detailed response to the comments. I suggest the manuscript can be accepted for publication in present form.

This manuscript is a resubmission of an earlier submission. The following is a list of the peer review reports and author responses from that submission.

Round 1

Reviewer 1 Report

Comments and Suggestions for Authors

Review of paper entitled "Large-scale production of anti-RNAse A VHH expressed in  pyrG auxotrophic Aspergillus oryzae"

Overview

The review focuses on the application pyrG (-) Aspergillus oryzae as a host system for the expression of anti-RNAse A VHH antibodies. The authors generated a pyrG auxotrophic (-) A. oryzae RIB40 strain for this purpose. Binding and functional screening was carried out to confirm the functionality of the expressed antibody.

Abstract: 

It was not clear why the need to use a pyrG (-) strain. Please add that statement. 

Introduction: 

Line 80 onwards, is the authors refereeing to S. cerevisiae or A. oryzae? 

The use of pyrG for improved expression of domain antibodies has been shown with e coli. This should be added to the introduction 

Pls see: Landberg J, Wright NR, Wulff T, Herrgård MJ, Nielsen AT. CRISPR interference of nucleotide biosynthesis improves production of a single-domain antibody in Escherichia coli. Biotechnol Bioeng. 2020 Dec;117(12):3835-3848. doi: 10.1002/bit.27536. Epub 2020 Aug 29. PMID: 32808670; PMCID: PMC7818426.

Line 213: dansitometrically - spelling

Line 237: After extensive washing – what was the extensive wash protocol?

Line 291: only uridine and uridine but – mistake?

Fig 3c: The image of the growth and non growth in tubes is not visible. Can the authors have an Od reading to substantiate the claims? 

Fig 6: The pulldown analysis, why was there no negative control. Pull down without VHH but only RNAse A. This will be important to show that the column is not binding to RNAse A.

Additionally, why didn’t the authors carry out a simple ELISA to show the binding?

Line 477: no degradation of the 477 secreted protein was observed at any stage. How was this concluded? Was there daily sampling done?

The comparison done by the authors with other systems based on shake flask and fermenters is rather confusing to readers. It would be best to summarize it in a table and make them comparable. Present them in g/L to have the best comparison as each examples are expressing at different volumes. Additionally, the authors did not consider the role of the antibody structure in this discussion. Some antibody sequences are less soluble, even for VHH antibodies. This can also be a factor to explain the difference in yield between the examples. As the authors only expressed 1 antibody clone, it would be better if the authors tone down the claims of the method and present it as generally a suitable alternative.

Reviewer 2 Report

Comments and Suggestions for Authors

This is an interesting paper aimed at testing and validating a new expression system for the production of nanobodies. The manuscript follows a straightforward and appropriate experimental design. The study's primary limitation is that they used a single nanobody to test the expression system. Still, this proof-of-principle study opens the door for future investigations and provides an alternative to the current E—coli-based expression system. Considering the emergence of nanobodies as therapeutic and research tools, the manuscript may attract the interest of many from different research fields.

Specific comments:

Nice introduction, but it mainly focused on nanobodies, their properties and advantages. It would have been nice to discuss the current methods used to produce nanobodies in more detail and the rationale (or the need) for testing the pyrG auxotrophic Aspergillus oryzae as another expression system.

The study suggests that pyrG auxotrophic Aspergillus oryzae is a suitable alternative for the production of nanobodies. Although this is an interesting and worthy aim, I would have liked to know why we should use this expression system instead of the typical E. coli-based system. How does it compare in terms of cost-effectiveness for comparable protein yield (no need here to provide details, but to inform the readers if this is comparable to other expression systems or even cheaper)? The authors reported obtaining 44 mg/L of pure nanobodies, which is quite impressive. Whether such yield can be obtained with other nanobodies is unclear as the yields are often nanobody-specific. This should also be addressed in the discussion to inform the readers.

The authors can produce pure VHH antibodies using their expression system. How it compares to the standard E. coli system regarding affinity for the protein of interest is unclear. This is a critical issue when choosing an expression system for an antibody. The authors provide some SPR data on the affinity of their preps, but there is no data on how it compares with other expression systems. If the authors do not have the data, they should at least comment on this issue in the discussion. This is not a critical issue as the authors obtained a binding affinity in the low nanomolar range, which is sufficient for different applications.

The authors describe some of the drawbacks of E. coli expression systems in the discussion. Yet, many drawbacks cited in the discussion may eventually be encountered in the A. oryzae system (such as variable yield, as discussed above, or inefficient folding) as more nanobodies are tested. Other drawbacks cited for the E.coli expression systems rarely apply to producing nanobodies (such as insoluble inclusion bodies, etc.). The authors should generally tone down their negatives on E. coli expression systems. It is unlikely that researchers will abandon the E.coli system for the A. oryzae system. However, I certainly can see the need to use this system for problematic nanobodies that are hard to produce in E. coli.

In my opinion, the IMAC issue is not a big deal. We rarely need to do more than one IMAC purification step. The authors used only one, but again, they tested only one nanobody. So again, I would tone done the discussion on this issue.

Round 2

Reviewer 1 Report

Comments and Suggestions for Authors

The authors have addressed all the major issues.